# Customer Lifetime Value Prediction with Uncertainty Estimation Using Monte Carlo Dropout

## Abstract

Accurately predicting customer Lifetime Value (LTV) is crucial for companies to optimize their revenue strategies. Traditional deep learning models for LTV prediction are effective but typically provide only point estimates and fail to capture model uncertainty in modeling user behaviors. To address this limitation, we propose a novel approach that enhances the architecture of purely neural network models by incorporating the Monte Carlo Dropout (MCD) framework. We benchmarked the proposed method using data from Player Unknown's Battlegrounds (PUBG) Mobile which is one of the most downloaded mobile games in the world, and demonstrated a substantial improvement in predictive Top 5% Mean Absolute Percentage Error compared to existing state-of-the-art methods. Additionally, our approach provides confidence metric as an extra dimension for performance evaluation across various neural network models, facilitating more informed business decisions.

## 1 Introduction

Customer Lifetime Value (LTV) is defined as the revenue generated by a customer over a specified time period $T$, where $T$ may vary depending on the specific business applications. Accurate prediction of LTV has become crucial for companies seeking to optimize their service and plan revenue strategies. For instance, early identification of customers' long-term purchasing potential allows for more precise targeted and customized service, thereby significantly increasing overall revenues.

Existing approaches to LTV prediction generally fall into two categories: conventional RFM-based (Recency, Frequency, and Monetary) statistical methods [1, 2, 3] and machine learning (ML)-based predictive models [4, 5, 6, 7]. ML-based methods formulate LTV prediction as a supervised-learning problem. They usually outperform RFM-based statistical models by making use of more user features. They are especially useful in scenarios where users do not have prior purchasing history, making RFM-based models inapplicable. Deep learning models have demonstrated as effective tools for predicting LTV. However, these models typically generate only single numerical point estimates, which fail to capture the model uncertainty when characterizing user behavior [8, 9, 10, 11]. In practice, customer purchasing behavior is influenced by numerous factors that may not be fully captured by model structures and parameters. As a result, model uncertainty is inherent in LTV prediction. Single-point predictions do not account for the uncertainty and may result in biased estimates [12, 13, 14]. Consequently, relying on single-point predictions for LTV in production can potentially compromise overall operating revenues and diminishing positive customer feedback.

To mitigate these risks, it is essential to complement the single-point prediction with additional statistical measures, such as the mean, variance, and distribution of the predictions. This approach enhances the accuracy of business decision-making processes and reduces the likelihood of potential financial losses [15, 16].

Submitted to 38th Conference on Neural Information Processing Systems (NeurIPS 2024). Do not distribute.

A common approach to addressing this issue in other research areas is incorporating traditional statistical modeling [17, 18, 19, 20]. For instance, integrating a Gaussian Process (GP) block at the end of DNNs can provide a distribution of forecasts along with other statistical information [21]. However, this method may introduce unacceptable time cost in runtime-sensitive applications, such as LTV prediction [22, 23, 24]. Gal et al. (2016) has proposed a theoretical framework that interprets dropout training in DNNs as approximate Bayesian inference in deep GPs, offering reduced computational time (see also [25]).

The challenges associated with LTV prediction can be summarized as follows: 1) Traditional deep learning models provide only single-point predictions offering limited information. [9, 26, 27, 28]. 2) Incorporating explicit components to capture model uncertainty, such as a Gaussian Process block, can provide valuable confidence estimates, but it incurs significant computational cost [29, 30, 31, 32, 33]. To address these limitations, we propose a novel approach that represents model uncertainty using stochastic dropout.

The contributions of this paper are summarized as follows:

1. To the best of our knowledge, it is the first LTV prediction model purely based on neural networks that provide uncertainty quantification without the need for additional modules.

2. The proposed framework demonstrates significant improvements across multiple LTV metrics on different DNN architectures.

3. The proposed framework provides confidence metric as an additional dimension of measurement for evaluating various LTV models (shows in Figure 1).

## 2  Methodology

Dropout training in DNNs can be framed as approximate Bayesian inference in deep Gaussian processes [15]. This approach enables researchers to obtain the quantification of model uncertainty in DNN predictions from a mathematically rigorous perspective, without modifying the backpropagation process. Equation 1 illustrates the concept of treating Monte Carlo Dropout (MCD) as a Bayesian approximation, where predictions are obtained by averaging the results of multiple network evaluations under stochastic dropout conditions [25].

$$\hat{y} = \frac{1}{T} \sum_{j=1}^{T} f\left(x; w \cdot d_j\right) \tag{1}$$

The parameters in Equation 1 are detailed as follows: $x$ represents the input features; $\hat{y}$ denotes the prediction output; $T$ is the number of Monte Carlo trials; $w$ represents the parameter weights; and $d_j$ is the dropout masks. The term $f\left(x; w \cdot d_j\right)$ refers to the network's output given input $x$ and parameter weights, with element-wise multiplication by dropout masks. The pseudo-code for sampling in LTV prediction tasks using the MCD method is outlined in Algorithm 1.

---
**Algorithm 1** Implementation of MCD method

---
**Input:** test data $D_{test}$ containing input features of $N$ samples, i.e., $D_{test} = \{x_1, x_2, ..., x_N\}$
**Output:** forecast with uncertainty measurement for $N$ samples
**for** $i = 1$ to $N$ **do**
    take an individual sample $x_i$
    **for** $j = 1$ to $T$ **do**
        perform a single forward pass with dropout mask $d_j$, obtaining $\hat{y}_{ij} = f\left(x_i; w \cdot d_j\right)$
    **end for**
    produce a result vector as $[\hat{y}_{i,1}, \hat{y}_{i,2}, \hat{y}_{i,3}, \ldots, \hat{y}_{i,T}]$ for sample $x_i$
    calculate the mean $\hat{y}_i$ and variance $\hat{\sigma}_i$ of the result vector $[\hat{y}_{i,1}, \hat{y}_{i,2}, \hat{y}_{i,3}, \ldots, \hat{y}_{i,T}]$
**end for**
**Return:** $\hat{Y} = [\hat{y}_1, \hat{y}_2, \ldots, \hat{y}_N], \hat{\sigma} = [\hat{\sigma}_1, \hat{\sigma}_2, \ldots, \hat{\sigma}_N]$ for the $N$ samples

---

Algorithm 1 illustrates the stochastic sampling process through multiple trials of random dropout. Specifically, $T$ forward passes are conducted for each of the $N$ samples drawn from the test set,

producing a set of predictions. Each sample $x_i$ ($1 \leq i \leq N$), from the test set yields a result vector of size $T$, denoted as $[\hat{y}_{i,1}, \hat{y}_{i,2}, \ldots, \hat{y}_{i,T}]$. Then the mean, and variance for each vector are calculated, facilitating the construction of forecasts with uncertainty quantification for individual input samples in the test set.

# 3 Experiments

In the previous section, we discussed the interpretation of dropout as a Bayesian approximation and provided a detailed breakdown of how to implement the MCD approach during the inference stage to extract uncertainties. In this section, we present several experiments to evaluate the proposed framework using the data from Player Unknown's Battlegrounds (PUBG) Mobile which is one of the most played online games in the world having over 1 billion downloads. Both standard metrics for LTV prediction and the proposed new metric using confidence intervals are presented. Due to space constraints, we refer the reader to the Appendix A for additional details on standard LTV prediction metrics.

## 3.1 Experimental Setup

The prediction task was to estimate the purchasing amount for the subsequent month after a new user has engaged with the game for one week, and then select potential top spenders for downstream tasks. The dataset utilized in this study covered approximately 3 million users. To evaluate the effectiveness and generalizability of the proposed approach, the MCD framework is implemented on two base models: the Multi-Layer Perceptron (MLP) and the Deep and Cross Network v2 (DCNv2) [34]. MLP was selected due to its simplicity and moderate performance, providing a suitable baseline, while DCNv2 was chosen for its demonstrated state-of-the-art performance in recommender systems and LTV prediction research. Both models were trained using Mean Squared Error (MSE) loss on the log-transformed purchasing amounts.

## 3.2 Main Results

The first metric for evaluating the proposed method is confidence assessment, which has not been addressed in previous LTV prediction studies. Deep learning models producing incorrect predictions due to model uncertainty is inevitable. It is preferable for incorrect predictions to be associated with low confidence and correct predictions with high confidence. We benchmarked the model's performance across various confidence levels.

The confidence assessment in this chapter is represented by the accuracy versus confidence plot. The confidence intervals (CIs) for a sample $x_i$ are computed using the following equation: $CI = \hat{y}_i \pm z \frac{\hat{\sigma}_i}{\sqrt{T}}$, where $z$ ($0 \leq z \leq 1$) is the confidence threshold. For $N$ samples, accuracy is defined as the proportion of samples of which the true label $y$ falls in the range of $CI$. Thus, accuracy varies against the confidence threshold $z$, which is shown in Figure 1. The x-axis represents the confidence threshold $z$, while the y-axis shows the accuracy **x%** of predictions.

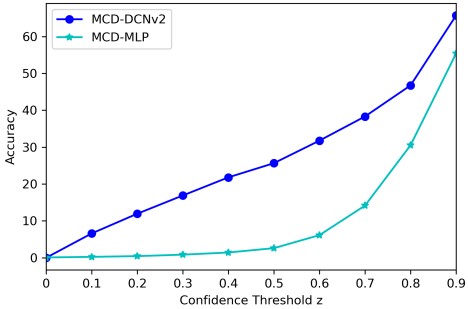

Figure 1: Model accuracy across different confidence intervals

As shown in Figure 1, the MCD-DCNv2 model exhibits higher accuracy across all confidence intervals. However, even at the optimal performance points of both models at $z = 0.9$, there remains a performance gap of 10%, with MCD-DCNv2 generating predictions with greater certainty.

Selecting the superior model for production needs to take into account more metrics which we present in the rest of this section. Three common metrics of LTV prediction were compared: normalized Gini coefficient[4], Mean Absolute Percentage Error (MAPE), and hit-rate (a.k.a., precision). Normalized Gini coefficient and hit-rate reflect a model's ability to generate the correct ranking of users. The former doesn't require a threshold similar to AUC, whereas the later needs a threshold which we set as top 5%. MAPE reflects how far the predicted values deviate from the actual values. It is usually applied to the entire data set for regression problems. However, our data labels are inflated with zeros, making MAPE infinity. Therefore, we computed MAPE for users in the top 5th percentile. Best performed MCD settings were selected for both MLP and DCNv2. We compared the proposed method with raw models, as well as the well-established Ziln loss method from LTV prediction literature introduced by Google [4]. Results are shown in Table 1.

| | Normalized Gini Coefficient | Top5% MAPE | Top5% Hit-Rate |
|---|---|---|---|
| MLP | 0.9605 | 0.4835 | 35.95% |
| MCD-MLP | **0.9638** | **0.1858** | 36.07% |
| DCNv2 | 0.9609 | 0.4226 | 36.12% |
| MCD-DCNv2 | 0.9637 | 0.2003 | **36.25%** |
| Ziln | 0.9581 | 0.7500 | 36.11% |

Table 1: Comparison of three common metrics for LTV prediction: normalized Gini coefficient (higher is better), MAPE (lower is better), and hit-rate (higher is better) across various model settings.

As presented in Table 1, the raw DCNv2 model outperforms the raw MLP model across all three metrics. However, the performance metrics shift after implementing the MCD framework. The proposed framework yields more substantial performance improvements in the MCD-MLP model compared to MCD-DCNv2, particularly in terms of the normalized Gini coefficient and top 5% MAPE. However, the MCD-DCNv2 model still achieves the highest top 5% hit rate.

Compared to Ziln, MCD-DCNv2 demonstrates superior performance, especially in MAPE metric. This advantage may be attributed to the fact that, although the Ziln loss method was designed to address the severe label imbalance issue in zero-inflated data for LTV prediction, it assumes a lognormal distribution for non-zero labels, which does not align with the distribution characteristics of our real-world datasets.

In business scenarios, models are evaluated holistically using multiple metrics and different models can be selected for production based on different priorities. First, the MCD framework offers significant performance enhancement, particularly within the MCD-MLP structure. Second, business stakeholders may prioritize a model that emphasizes robustness in confidence estimation to hedge risks, even at the expense of less performance gain in predicted values. In this case, MCD-DCNv2 is preferred.

## 4   Conclusion and Future Work

We proposed the MCD framework that represents the first application of a purely neural network-based prediction model in the LTV domain that incorporates uncertainty measurement without introducing additional modules. Experiments on a real-world dataset were conducted, illustrating that the proposed framework also improved the conventional metrics. We advocate the usage of uncertainty assessment for LTV applications to support more informed and reliable decision-making in business contexts.

In future work, we aim to investigate the potential of incorporating uncertainty measures as features in user segmentation tasks, such as identifying high-value users in gaming contexts [7, 35, 36, 37]. We also plan to benchmark the proposed MCD framework on more DNN architectures and investigate other uncertainty-aware models, such as deep ensembles [22, 38, 39] originally introduced by Google DeepMind.

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

# A    Appendix: Supplementary Figures

In the previous results section, the optimal prediction outcomes generated by various model architectures were compared. We also investigated the impact of hyperparameter settings on the predictive performance of MCD models. We visualized the change in normalized Gini coefficient [4] and Top 5% MAPE as the number of MCD trials increased in Figure 2. The performance of the raw models is also included as a baseline for comparison.

Figure 2(a) presents the performance on normalized Gini coefficient. Raw DCNv2 (dash-dot line) and raw MLP (dotted line) models are shown as baseline references. Raw DCNv2 model outperforms the raw MLP model. However, following the integration of MCD into these models, MCD-MLP model achieves a higher normalized Gini improvement compared to the MCD-DCNv2 model. Furthermore, normalized Gini coefficients for both MCD models show a trend toward convergence as the MCD trial size increases.

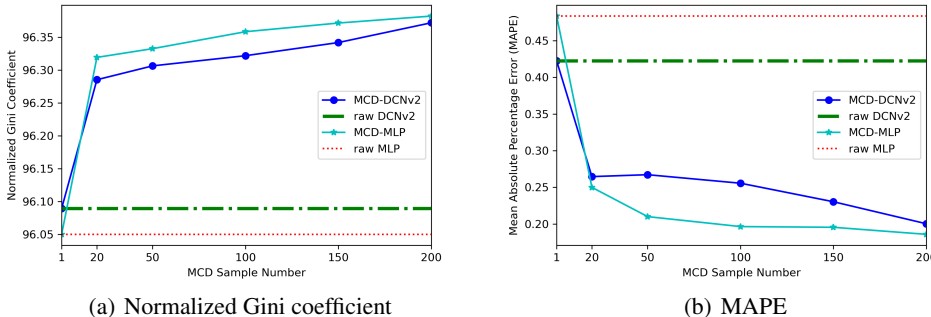

(a) Normalized Gini coefficient                    (b) MAPE

Figure 2: Major metrics using different MCD trials. Both the normalized Gini coefficient and MAPE improved as the number of trials increased.

Figure 2(b) shows the performance on Top 5% MAPE. A lower MAPE value indicates that a model produces more accurate predictions. The Top 5% MAPE performance of the raw DCNv2 (dash-dot line) and raw MLP (dotted line) models is presented as baselines. It can be observed that implementation of the MCD framework results in an approximately 50% improvement in MAPE performance for both raw models. Additionally, consistent with the observed gains in the normalized Gini coefficient, the MCD-MLP model demonstrates a more substantial performance improvement compared to the MCD-DCNv2 model, even though the raw DCNv2 model initially outperforms the raw MLP model. Moreover, both MCD models exhibit a trend toward convergence in MAPE performance as the MCD sample size increases.

One downstream application for LTV prediction is to design user acquisition marketing strategies. Gini and MAPE performance gains are not always the primary concerns for business owners. When market conditions decline and campaign funding is constrained, the reliability of a model and its ability to minimize uncertainties become critical factors in business decision-making. Therefore, considering the results shown in Figures 1 and 2, the MCD-DCNv2 model, compared to MCD-MLP, offers significantly higher confidence in its predictions, at the cost of only a 0.01% reduction in Gini and a 1.45% decrease in MAPE performance. This trade-off is straightforward and advantageous for consideration in business applications.

