# OpenReview forum: "Customer Lifetime Value Prediction with Uncertainty Estimation Using Monte Carlo Dropout"
_NeurIPS.cc/2024/Workshop/BDU — Submitted to NeurIPS BDU Workshop 2024_

### Official Review · Reviewer_Z72d · 2024-09-22
**need more to improve**

**Rating:** 3
**Confidence:** 5

**Review:**

**Pros:**
1. **Innovation**: The paper presents a novel approach by incorporating the Monte Carlo Dropout framework to quantify uncertainty in neural network models, which is a relatively new application in customer lifetime value (LTV) prediction.
---
**Cons:**
1. **Loose structure**: The overall structure of the paper is disjointed, with weak transitions between sections, making it difficult for readers to follow the research flow.
2. **Insufficient experimental details**: The experimental section lacks sufficient details on model training settings and data preprocessing, making it difficult to replicate the experiments.
3. **Weak explanation of methodology**: The theoretical background of the Monte Carlo Dropout method is not adequately explained, and its unique advantages in LTV prediction are not thoroughly addressed.
4. **Lack of comparative experiments**: The paper does not include sufficient comparisons with other mainstream methods, especially lacking consideration of more complex neural network models or ensemble learning approaches.
5. **Over-reliance on a single dataset**: The paper relies on a single dataset (PUBG Mobile), which limits the demonstration of the method's generalizability across other domains or datasets.
6. **Narrow selection of evaluation metrics**: Although several metrics are employed, there is insufficient discussion on whether these metrics are appropriate for evaluating the actual business impact of LTV prediction models.
7. **Inadequate discussion**: The analysis of the experimental results is superficial, failing to delve into the potential impact of uncertainty in model predictions on business decision-making.
8. **Poor language clarity**: Some paragraphs are overly verbose and repetitive, diluting the clarity of the arguments and making it harder to convey the research contributions effectively.
---
**Review Conclusion:**
While the paper proposes a novel method, it fails to adequately explain the theoretical basis and experimental details. The innovation is commendable, but significant flaws in the experimental design and result analysis undermine the credibility and academic value of the work. It is strongly recommended that the authors improve the experimental design, include more comparative studies, and strengthen the theoretical explanations and discussions of the results to enhance the overall quality of the paper.

---

### Official Review · Reviewer_u7eX · 2024-10-06
**The paper introduces Monte Carlo Dropout into neural network models for Customer Lifetime Value prediction to estimate uncertainty and improve predictive accuracy.**

**Rating:** 7
**Confidence:** 4

**Review:**

The paper

(i) proposes a novel approach to Customer Lifetime Value (LTV) prediction by incorporating Monte Carlo Dropout (MCD) into neural network models to estimate uncertainty. Traditional deep learning models for LTV prediction typically provide point estimates and fail to capture model uncertainty, which is crucial for informed business decisions. By integrating MCD, the authors enable the models to produce both predictions and uncertainty estimates without significant computational overhead. They validate their approach using data from PUBG Mobile, demonstrating substantial improvements in predictive accuracy and the introduction of confidence metrics as an extra dimension for performance evaluation.

(ii) focuses on MCD but does not thoroughly compare its performance with other uncertainty estimation techniques. Including such comparisons would strengthen the validation of the approach.

(iii) could benefit from a deeper theoretical discussion on why MCD improves LTV prediction and how uncertainty quantification impacts model performance and business outcomes.

---

### Decision · Program_Chairs · 2024-10-09

**Decision:**

Reject

**Comment:**

Reviews for this paper are mixed, with one positive and one negative. The negative review complains about lack of comprehensiveness in terms of a whole host of aspects such as baselines, metrics, choices of datasets, and other aspects. The positive review complains about the same issues but provides a much higher score, perhaps due to valuing the authors' application. My overall impression is that, due to the relatively large number of distinct evaluation issues raised, this work could use more time and another refereeing round before it becomes ready to present. I therefore encourage the authors to refine the work, improve those aspects, resubmit to a future workshop.